# Perihematomal Edema and Clinical Outcome in Intracerebral Hemorrhage Related to Different Oral Anticoagulants

**DOI:** 10.3390/jcm10112234

**Published:** 2021-05-21

**Authors:** Jawed Nawabi, Sarah Elsayed, Andrea Morotti, Anna Speth, Melanie Liu, Helge Kniep, Rosalie McDonough, Gabriel Broocks, Tobias Faizy, Elif Can, Peter B. Sporns, Jens Fiehler, Bernd Hamm, Tobias Penzkofer, Georg Bohner, Frieder Schlunk, Uta Hanning

**Affiliations:** 1Department of Radiology, Charité-Universitätsmedizin Berlin, Campus Mitte, Humboldt-Universität zu Berlin, 10117 Berlin, Germany; e.can@charite.de (E.C.); b.hamm@charite.de (B.H.); t.penzkofer@charite.de (T.P.); 2BIH Biomedical Innovation Academy, Berlin Institute of Health (BIH), 10178 Berlin, Germany; f.schlunk@charite.de; 3Department of Diagnostic and Interventional Neuroradiology, University Medical Center Hamburg Eppendorf, 20246 Hamburg, Germany; s.elsayed@uke.de (S.E.); h.kniep@uke.de (H.K.); rosevmcd@gmail.com (R.M.); g.broocks@uke.de (G.B.); peter.sporns@hotmail.de (P.B.S.); Fiehler@uke.de (J.F.); u.hanning@uke.de (U.H.); 4Neurology Unit, Department of Clinical and Experimental Sciences, University of Brescia, 25123 Brescia, Italy; andrea.morotti85@gmail.com; 5Department of Neuroradiology, Charité-Universitätsmedizin Berlin, Campus Mitte, Humboldt-Universität zu Berlin, Berlin Institute of Health, Freie Universität Berlin, 10117 Berlin, Germany; anna.speth@charite.de (A.S.); melanie.liu@charite.de (M.L.); g.bohner@charite.de (G.B.); 6Department of Radiology, Stanford University School of Medicine, Stanford, CA 94305, USA; tfaizy@stanford.edu; 7Department of Neuroradiology, Clinic for Radiology and Nuclear Medicine, University Hospital Basel, 4031 Basel, Switzerland

**Keywords:** edema, anticoagulation, cerebral hemorrhage, computed tomography, outcome prediction

## Abstract

Background: There is a need to examine the effects of different types of oral anticoagulant-associated intracerebral hemorrhage (OAC-ICH) on perihematomal edema (PHE), which is gaining considerable appeal as a biomarker for secondary brain injury and clinical outcome. Methods: In a large multicenter approach, computed tomography-derived imaging markers for PHE (absolute PHE, relative PHE (rPHE), edema expansion distance (EED)) were calculated for patients with OAC-ICH and NON-OAC-ICH. Exploratory analysis for non-vitamin-K-antagonist OAC (NOAC) and vitamin-K-antagonists (VKA) was performed. The predictive performance of logistic regression models, employing predictors of poor functional outcome (modified Rankin scale 4–6), was explored. Results: Of 811 retrospectively enrolled patients, 212 (26.14%) had an OAC-ICH. Mean rPHE and mean EED were significantly lower in patients with OAC-ICH compared to NON-OAC-ICH, *p*-value 0.001 and 0.007; whereas, mean absolute PHE did not differ, *p*-value 0.091. Mean EED was also significantly lower in NOAC compared to NON-OAC-ICH, *p*-value 0.05. Absolute PHE was an independent predictor of poor clinical outcome in NON-OAC-ICH (OR 1.02; 95%CI 1.002–1.028; *p*-value 0.027), but not in OAC-ICH (*p*-value 0.45). Conclusion: Quantitative markers of early PHE (rPHE and EED) were lower in patients with OAC-ICH compared to those with NON-OAC-ICH, with significantly lower levels of EED in NOAC compared to NON-OAC-ICH. Increase of early PHE volume did not increase the likelihood of poor outcome in OAC-ICH, but was independently associated with poor outcome in NON-OAC-ICH. The results underline the importance of etiology-specific treatment strategies. Further prospective studies are needed.

## 1. Introduction

In light of the aging population with increased cardiovascular comorbidity, the use of oral anticoagulation (OAC) is steadily expanding [1,2]. The incidence of oral anticoagulation-related intracerebral hemorrhage (OAC-ICH) is growing, due to the increasing variety of medical treatment options [1,3]. The optimal treatment strategy is still uncertain, and prognosis in OAC-ICH is often associated with greater morbidity and mortality compared to non-oral anticoagulation-related intracerebral hemorrhage (NON-OAC-ICH) [1,2,3,4]. Over the temporal course of the initial insult, formation of perihematomal edema (PHE) may develop, expressing secondary brain injury [5]. There is evidence from both clinical and experimental studies to suggest that in the context of OAC, early PHE formation is altered [6,7,8]. In light of novel alternative OAC treatment options, the generalizability of these results needs to be validated, as these findings were restricted to vitamin-K-antagonist [1,9]. At the same time, PHE is gaining increasing attention as a promising surrogate marker not only for secondary brain injury, but also clinical outcome [10,11,12,13]. In light of these reports, a better understanding of PHE formation in OAC-ICH patients could ultimately help to improve clinical treatment, when utilizing established PHE quantification methods. We hypothesized (1) lower early PHE in patients with OAC-ICH compared to NON-OAC-ICH (2) and therefore a less predictive value for clinical outcome. To test and evaluate this hypothesis, we present a two-phase analysis: First, computed tomography (CT)-derived imaging markers for early PHE were calculated in patients with OAC-ICH and NON-OAC-ICH (absolute PHE, relative PHE (rPHE), edema expansion distance (EED)). A further subgroup analysis for differences between NOAC and vitamin K antagonists (VKA) was performed. In a second approach, a logistic regression model was established to identify differences in independent predictors of clinical outcome in OAC-ICH and NON-OAC-ICH.

## 2. Material and Methods

### 2.1. Study Population

We retrospectively parsed the databases of two German tertiary stroke centers for patients with spontaneous ICH aged >18 years between January 2016 and April 2019. (University Medical Center Hamburg-Eppendorf, Germany; and Charité University Hospital Berlin, Germany). As inclusion criteria, we defined (1) primary acute ICH confirmed on NCCT (Non-contrast Computed Tomography), with or without CT angiography (CTA), (2) with a symptom onset within 12 h. Both databases excluded patients with head trauma, brain tumor, vascular malformation, primary intraventricular hemorrhage, or secondary ICH from hemorrhagic transformation of ischemic infarction. Clinical data for OAC and antiplatelet therapy were documented. Type of OAC medication was documented if available. Clinical parameters also included vascular risk factors (arterial hypertension and diabetes mellitus), time difference from symptom onset to NCCT, both Glasgow Come Scale (GCS) and National Institutes of Health Stroke Scale (NIHSS) on admission, clinical outcome defined by modified Rankin scale (mRS) at 90 days, and surgical procedures (craniectomy, extra-ventricular drainage (EVD) placement) from patients’ clinical records were additionally obtained. Patients were dichotomized into patients with OAC-ICH and NON-OAC-ICH. This multicenter retrospective study was approved by the ethics committee (Ethik-Kommission der Ärztekammer Hamburg, Ethik-Kommission der Charité Berlin) and written informed consent was waived by the institutional review boards. All study protocols and procedures were conducted in accordance with the Declaration of Helsinki. Patient consent was not needed (because of the retrospective nature of the study). The data that support the findings of this study are available from the corresponding authors upon reasonable request.

### 2.2. Imaging

#### 2.2.1. Image Acquisitions

NCCT scans were performed using standard clinical parameters, with an axial <5 mm section thickness. All datasets were inspected for quality and excluded in case of severe motion artifacts. In detail the images were acquired on the following scanners: 256 slice scanner (Philips iCT 256, Philips, Amsterdam, Netherlands) with 120 kV, 280–320 mA, <5.0 mm slice reconstruction and <0.5 mm in-plane resolution and CTA with 100–120 kV, 260–300 mA, 1.0 mm slice reconstruction, 5 mm MIP (maximum intensity projection) reconstruction with 1 mm increment, 0.6-mm collimation, 0.8 pitch, H20f soft kernel, 80 mL highly iodinated contrast medium and 50 mL NaCl flush at 4 mL/s; scan starts 6 s after bolus tracking at the level of the ascending aorta. Eighty-slice scanner (Toshiba Aquilion Prime, Toshiba, Tokyo, Japan) with 120 kV, 280 mA, <5.0 mm slice reconstruction and <0.5 mm in-plane resolution and CTA with 100–120 kV, 260–300 mA, 1.0 mm slice reconstruction, 5 mm MIP reconstruction with 1 mm increment, 0.5-mm collimation, 0.8 pitch, H20f soft kernel, 60 mL highly iodinated contrast medium, and 30 mL NaCl flush at 4 mL/s; scan starts 6 s after bolus tracking at the level of the ascending aorta.

#### 2.2.2. Image Analysis

Data were retrieved in Digital Imaging and Communications in Medicine (DICOM) format from the local picture archiving and communication system (PACS) servers and anonymized in compliance with the local guidelines. Two experienced neuroradiologists (JN and SE) assessed and documented the following imaging features on admission and follow-up NCCT scans: (1) intraventricular hemorrhage; (2) ICH location; (3) craniectomy or EVD placement in the follow-up CCT scans. ICH locations were classified as basal ganglia, thalamus, lobar, brainstem/pons, and cerebellar. In the following process, ICH and PHE were segmented semi-automatically on the basis of the original NCCT images [14,15]. Regions of interest (ROIs) were delineated using Analyze 11.0 Software and ITK-SNAP 3.8.0 Software (University of Pennsylvania, Philadelphia, PA, USA and University of Utah, Salt Lake City, UT, USA) [15,16,17]. The ROI histogram for ICH was sampled between 20 and 80 Hounsfield units (HU) to exclude voxels that likely belong to cerebrospinal fluid or calcification. The ROI histogram for PHE was sampled between 0 and 30 HU to exclude voxels that likely belong to leucariosis [14]. Consensus ROIs were derived based on overlapping segmentations of both readers. Both readers were blinded to all clinical information and bleeding location. Discrepancies were settled by joint discussion of the 2 readers and a third and fourth reader, FS and UH. (JN, SE, and FS: 4 years clinical experience in diagnostic neuroradiology in an academic full-service hospital; UH: 8 years clinical experience in diagnostic neuroradiology; research with focus on clinical applications of image processing and predictive modelling; JN, SE, FS, and UH: Research with focus on clinical applications of image processing and predictive modelling).

#### 2.2.3. Perihematomal Edema Measurements

Studies evaluating PHE have used several varying parameters and definitions to assess PHE [18]. These studies also exhibited variabilities in the timing (single time point vs. peak) and method of assessing PHE progression (absolute increase vs. percent change vs. rate or speed). The various parameters used in this study were based on markers described for NCCT admission imaging (Figure 1):PHE _ABSOLUTE_: Refers to the absolute perihematomal edema volume on admission (PHE) [15,19].PHE _RELATIVE_: Relative perihematomal edema volume (rPHE) refers to the ratio absolute perihematomal edema volume (PHE) compared to absolute ICH volume [3,11,20,21].Edema Extension Distance: Edema extension distance (EED) refers to the difference between the radius of a sphere equal to the absolute PHE volume and the radius of a sphere equal to the ICH volume alone. In brief, PHE volume corresponds to the absolute edema volume and π corresponds to the radius of a sphere equal to the combined volume of PHE and ICH and π the radius of a sphere equal to the volume of the ICH alone [18,22]. This can be calculated using the following formula:
$$EED=\sqrt[{3}]{\frac{PHEvol+ICHvol}{\frac{4}{3}\pi }}-\sqrt[{3}]{\frac{ICHvol}{\frac{4}{3}\pi }}$$

### 2.3. Clinical Outcomes

The primary outcome was poor outcome, defined by the modified Rankin scale (mRS) at 90 days. The mRS 90 was analyzed as a dichotomous variable, as this has been the standard in ICH clinical trials, and defined as mRS 0–3 being a good and mRS 4–6 a poor outcome [23,24,25]. Raters (JN and SE) were trained in the use of the mRS 90 and blinded to each other, imaging, and non-relevant clinical data.

### 2.4. Statistical Methods

Data were tested for normality and homogeneity of variance using histogram plots and a Shapiro-Wilk test. Descriptive statistics are presented as counts (percentages (%)) for categorical variables, mean (standard deviation (SD)) for continuous normally distributed variables, and medians (interquartile range (IQR)) for non-normal continuous variables. Unadjusted differences in baseline and imaging characteristics (NON-OAC-ICH versus OAC-ICH) were evaluated using the Fisher exact test (2-tailed), Kruskal-Wallis test, or unpaired t test, as appropriate. A statistically significant difference was accepted at a *p*-value of less than 0.05.

#### 2.4.1. Exploratory Analyses

An exploratory approach was conducted for patients with a documented type of OAC medication. A one-way analysis of variance (ANOVA) was conducted to evaluate differences in PHE formation in patients within the three groups: (1) NON-OAC-ICH, (2) OAC-ICH with NOAC, (3) and OAC-ICH with VKA. The assumption of homogeneity of variances was tested using Levene’s test. Post hoc comparisons using the Tukey HSD test was selected in case of a significant effect for the overall ANOVA. A statistically significant difference was accepted at a *p*-value less than 0.05.

#### 2.4.2. Regression Analyses

We performed a univariable and multivariable logistic regression analyses to identify covariates associated with poor outcome (mRS 4–6 at 90 days) in patients with OAC-ICH and NON-OAC-ICH. Multivariable model building proceeded as follows: first, covariates with *p* < 0.1 in univariable analyses were included; second, universal confounders (age and sex) were force entered; third, covariates with *p* > 0.1 were backward eliminated; fourth, collinear covariates, as expressed by a variance inflation factor (VIF) > 3, were identified, and 1 covariate was removed from the model [26]. Specific location (lobar versus deep) was included as a covariate. For all statistical analyses, a 2-sided *p* of 0.05 was set as the significance threshold, and 95% CIs (confidence intervals) were reported for all odds ratios.

Statistical analyses were performed using the IBM SPSS Statistics 21 software package (IBM Corporation, Armonk, NY, USA). Missing data regarding basic characteristics, neuroimaging, or outcome led to exclusion of patients (Figure 2).

## 3. Results

### 3.1. Clinical Parameters

Our analysis included NCCT images of 811 patients with acute primary ICH who fulfilled the inclusion criteria. A total of 212 (26.14%) patients were grouped as OAC-ICH and 599 (73.86%) as NON-OAC-ICH. A detailed patient flowchart is given in Figure 2. Patients with OAC-ICH were significantly older, with a median age of 77 years (IQR 70–82) compared to a median age of 70 years (IQR 58–77) in patients with NON-OAC-ICH, *p*-value < 0.001. There were no differences in sex between both groups, *p*-value 0.61. Patients with OAC-ICH had a higher percentage of arterial hypertension, with 83.49% in 177 patients and diabetes mellitus with 18.87% in 40 patients compared to patients with NON-OAC-ICH, *p*-value 0.008 and 0.048, respectively. Use of antiplatelet medications was higher in patients with NON-OAC-ICH, with 26.7% in 160 patients compared to 14.15% in 30 patients with OAC-ICH, *p*-value < 0.001. Median time from symptom onset to imaging was almost similar between both groups, with 2.74 (IQR 1.56–12.65) h in patients with OAC-ICH compared to 2.81(IQR 1.28–13.57) h in patients with NON-OAC-ICH, *p*-value 0.592. Both GCS and NIHSS on admission were not statistically different between both groups, with a GCS of 11 (IQR 4–14) and NIHSS of 6 (IQR 1–14) in OAC-ICH, and a GCS of 12 (IQR 5–14) and NIHSS of 8 (IQR 1–15) in NON-OAC-ICH, *p*-value 0.315 and 0.44, respectively.

### 3.2. Surgical Procedures and Clinical Outcome

Surgical procedures including supratentorial or suboccipital craniectomy or EVD placement did not differ in both groups (details displayed Table 1), *p*-value 0.358, 0.85 and 0.785, respectively. Dichotomized clinical outcome with mRS 0–3 and 4–6 were similar between both groups, with 61 patients (28.77%) in OAC-ICH and 164 (27.38%) in NON-OAC-ICH for mRS 0-3, *p*-value 0.24. A total of 151 (71.23%) patients in OAC-ICH and 435 (72.62%) patients in NON-OAC-ICH had a mRS of 4–6, *p*-value 0.7. There was no significant difference in mortality (mRS 6) between both groups, as 147 patients (24.54%) with NON-OAC-ICH and 58 (27.35%) with OAC-ICH died, *p*-value 0.068.

### 3.3. Radiological Parameters

Absolute ICH volume and absolute PHE volume did not differ between both groups, *p*-value 0.767 and 0.091, respectively. Both rPHE and EED were significantly lower in patients with OAC-ICH, with a mean rPHE of 1.08 (SD 2.09) and a mean EED of 4.14 cm (SD 2.22) compared to a mean rPHE of 1.25 (SD 2.5) and a mean EED of 4.72 cm (SD 2.58) in patients with NON-OAC-ICH, *p*-value 0.001 and 0.007, respectively. Intraventricular hemorrhage (IVH) was almost similar, with 44.81% in OAC-ICH and 46.58% in NON-OAC-ICH, *p*-value 0.799. ICH location in the basal ganglia was higher in patients with NON-OAC-ICH, with 39.7% compared to 32.1% in OAC-ICH, *p*-value 0.05. There were no statistically significant differences regarding the other ICH locations (Table 1).

### 3.4. Exploratory Analyses 

In an exploratory analysis, differences in PHE formation depending on the type of oral anticoagulation medication were analyzed. From the 811 included patients, the type of OAC medication was documented for NOAC and VKA in 174 cases (29.05%). NOAC included dabigatran, rivaroxaban, and apixaban. VKA included phenprocoumon as their pharmaceutical agent.

A one-way analysis of variance was conducted to evaluate differences in PHE formation in patients within the following three groups: (1) NON-OAC-ICH (*n* = 599), (2) OAC-ICH with NOAC (*n* = 93), (3) and OAC-ICH with VKA (*n* = 81). Results for independent variables are displayed in Table 2. Since the ANOVA was significant for EED at a *p*-value < 0.05 level for the three groups, a post hoc test was computed. The Tukey HSD test indicated that the mean score for EED in OAC-ICH with NOAC was significantly lower than the EED in NON-OAC-ICH (Mean 0.65, SD 0.28, 95% CI 0.01–1.31, *p*-value 0.05). There was no statistical difference for EED between OAC-ICH with NOAC and VKA (Mean 0.27, SD 0.38, 95% CI −0.62–1.17, *p*-value 0.76).

### 3.5. Perihematomal Edema Based Clinical Outcome Prediction

Univariate and multivariate logistic regression analyses were performed in a separate approach for OAC-ICH and NON-OAC-ICH to analyze predictors of poor outcome (mRS 4–6 at 90 days), with special regard to individual prognostic effects of PHE. Independent variables included age; sex; arterial hypertension; diabetes mellitus; antiplatelet medication; time from symptom onset to imaging; both GCS and NIHSS on admission, both absolute ICH and PHE volume; rPHE, EED, IVH, ICH location; and craniectomy (Appendix A). The remaining independent variables in multivariate model for (1) OAC-ICH were NIHSS and GCS (Table 3). Lower GCS significantly increased the likelihood of poor outcome (odds ratio (OR) 0.76 for 1-point increase; 95% CI 0.63–0.91; *p*-value = 0.003), and a higher NIHSS increased the likelihood for poor outcome (per 1-point increase; OR 1.62; 95% CI 1.02–1.22; *p*-value 0.10). (2) The remaining independent variables in multivariate model for NON-OAC-ICH were sex, GCS, NIHSS, PHE volume, EED, IVH, and ICH location (Table 4). Female sex significantly decreased the likelihood of poor outcome (odds ratio (OR) 0.52; 95% CI 00.283–0.955; *p*-value = 0.035). Lower GCS significantly increased the likelihood of poor outcome (OR 0.76 for 1-point increase; 95% CI 0.68–0.85; *p*-value < 0.0001), and a higher NIHSS significantly increased the likelihood of poor outcome (per 1-point increase; OR 1.14; 95% CI 1.08–1.20; *p*-value < 0.001). Higher absolute PHE volume significantly increased the likelihood of poor outcome (OR 1.01 for 1 mL increase; 95% CI 1.00–1.02; *p*-value < 0.027). Higher EED did not increase the likelihood of poor outcome; *p*-value < 0.843). Presence of IVH significantly increased the likelihood of poor outcome (OR 2.8; 95% CI 1.44–5.45; *p*-value < 0.002), and presence of supratentorial ICH significantly decreased the likelihood of poor outcome (OR 0.38; 95% CI 0.20–0.72; *p*-value < 0.003). ICH volume was excluded from the multivariate regression model in both groups as a strong collinear covariate of PHE volume, with a VIF > 5.

## 4. Discussion

In this study, we analyzed differences of early PHE formation in patients with OAC-ICH in comparison with NON-OAC-ICH, and in particular related to different types of OAC treatment options. The main finding of our study is that quantitative markers of early PHE are significantly lower in patients with OAC-ICH compared to NON-OAC-ICH. To further elucidate potential differences in PHE formation within different types of OAC, we analyzed quantitative markers of PHE in NOAC and VKA associated OAC-ICH in comparison with NON-OAC-ICH. A conclusive secondary finding of this study was that early PHE volumes were not independently associated with poor outcome in OAC-ICH, but were significantly associated with poor outcome in patients with NON-OAC-ICH.

To our knowledge, this is the first study to add findings on PHE formation in NOAC- and VKA-associated ICH. In our study, mean EED levels were significantly lower in NOAC-ICH compared to NON-OAC-ICH. However, the actual difference in the mean scores between groups was quite small, based on Cohen’s conventions for interpreting effect size; as the type of oral anticoagulation medication was not documented in all cases [27]. Nevertheless, it is believed that within a larger effect size other parameters of PHE formation, i.e., rPHE, may have also differed between the groups. These results may be attributed to the biochemical properties of the NOACs and their differences from the mechanism of action of VKAs. Thrombin in particular has been identified as a potent stimulator and key link for early PHE formation and therefore might be inhibited to a stronger level in NOAC than in VKA [28,29,30] and hence contribute to a stronger attenuation of PHE formation.

Each of the proposed PHE parameters in our study has advantages and limitations. PHE is strongly related to the size of the underlying ICH and, therefore, alone may not account for this intermixed relationship [18]. rPHE can be disproportionally large in a smaller ICH, which may render it unsuitable for examining the relationship with outcome in some cases [18,20,31]. EED is independent of the concomitant influence of ICH volume on PHE and therefore has major advantages from a clinical trial perspective [22,32]. In line with this, EED may be capable of discerning the relationship of PHE with OAC and NON-OAC-ICH, especially those treated with NOACs.

The use of NOAC is expected to increase as randomized controlled trials provide solid evidence for the favorable risk–benefit profile of NOACs compared to VKAs [9,33,34]. In addition, the expected approval of different reversal agents will further increase its use [34,35]. With lower levels of EED in NOACs compared to NON-OAC-ICH, their use might further increase the safety of OAC, as the edema’s impairment might be less. A better understanding of the pathophysiology of NOAC and VKA-related ICH may allow us to further adapt treatment regimens. In this sense, patients with NON-OAC-ICH might beneficially profit from innovative and promising medications targeting early PHE, whereas patients with OAC-ICH might benefit primarily from potential treatments for the cessation or control of hematoma volume and expansion [1,3]. Such ongoing research into new, alternative treatment regimes in ICH remains crucial, as current treatment options have failed to provide promising, and ultimate, improvements in functional outcome and mortality. Increasing studies have therefore addressed PHE formation as a potential new therapy target. Animal experimental studies demonstrated that fingolimod could reduce PHE formation, cell apoptosis, and cerebral atrophy following ICH [36]. Similar results were observed in a clinical study [37]. Likewise, dexamethasone was shown to reduce cerebral cell apoptosis and inhibit brain inflammation [38]. However, discrimination between patients with OAC and non-OAC was omitted and may have been a potential limiting factor in previous randomized controlled trials (RCT) in further studying the treatment effects of antiedematous drugs in ICH: a RCT with 128 supratentorial ICH patients observed that mannitol failed to significantly improve the outcome at the end of 30 days and decrease mortality [39]. Our study adds preliminary conclusions that patients with NOACs may be less affected by early PHE formation. This conclusion does not only offer interesting pathophysiological insights into the potentially different impact on thrombin formation of new OAC in comparison to VKA-related ICH, but underlines the importance of etiology-specific treatment strategies in ICH. The conclusions of our study are clearly limited to the early phase of PHE formation. Nevertheless, medical treatment for secondary inflammation and reducing PHE only has a short-of-effective time window and is most effective if instituted early [40], as clinical studies in humans have suggested rapid PHE growth within the first 24 h following ICH [10]. Future clinical studies should therefore also elucidate the dynamics of PHE over time to tailor etiology-specific treatment strategies in patients with ICH. Such treatment strategies could be stratified in future RCT and could include early antidot treatment for OAC and anti-ICH expansion treatments in patients with OAC; and on the other hand the inclusion of antiedematous drugs in patients with NON-OAC-ICH at an early stage of diagnosis, monitored by quantitative assessment of PHE via EED as a surrogate marker.

As ICH volumes in OAC-ICH tend to be initially larger and expand more extensively, we therefore assume that the detrimental effects of a larger ICH volume in OAC-ICH by far outweigh the potential protective effects of thrombin deficiency. Figure 1 shows an illustrative example of a patient with OAC-ICH and the fluid levels, a marker for coagulopathy and hematoma expansion (HE) [41]. Follow-up CT in this patient revealed a massive hematoma expansion with IVH, although acute PHE on the admission NCCT was comparably low.

Although OAC has been described as a clear independent risk factor of major bleeding events, the distribution of poor outcome and mortality did not differ in both cohort groups in our study. In clinical trials of novel oral anticoagulants, namely the oral direct thrombin inhibitors and factor Xa inhibitors, major bleeding rates were generally low and comparable to those with LMWH or VKA-related ICH [42,43]. A meta-analysis examined the safety and efficacy of novel oral anticoagulants compared with warfarin for the prevention of stroke and systemic embolism in atrial fibrillation, and ICH was reduced in patients receiving novel oral anticoagulants (RR 0.49, 95% CI 0.36–0.66) [44]. In addition, six good-quality RCTs compared NOACs (2 DTI studies, 4 FXa inhibitor studies) with Warfarin. In patients with atrial fibrillation, NOACs decreased all-cause mortality, including fatal hemorrhages [43]. When looking at our patient cohort, a large proportion of patients in our study cohort were under medication with NOACs (*n* = 93) in comparison to patients treated with VKA (Marcumar; *n* = 81). These findings may be explanatory for the non-significant differences of poor outcome and mortality between the mRS of patients with NOAC-ICH and OAC-ICH.

The strengths of our study were the large sample size and the use of two multicenter cohorts. Nevertheless, our study has several limitations. First, we lacked clinical data on the medication of therapeutic options available for OAC reversal (i.e., NOAC antidote) and hemostatic therapy [39]. Patients with OAC-ICH were significantly older and displayed a higher percentage of arterial hypertension, which are both known to be common in this patient subset [1,3]. These factors might have contributed to poor clinical outcome and limit the generalizability of our results. However, arterial hypertension and age were no significant variables in the multivariate regression analysis. Patients with a symptom onset greater than three hours were included, arguably mitigating the effect OAC had on solely the early stage of PHE formation, yet median time from symptom onset was below three hours in both groups [13]. We assessed PHE volume at baseline, from the first CT scan. As follow-up imaging was not available for all patients, the measurement of change in PHE volume over time was not possible. Rate of change in PHE volume could be associated with functional outcome independently of ICH growth rate, and our present study was unable to determine this.

## 5. Conclusions

Quantitative markers of early PHE (rPHE and EED) were lower in patients with OAC-ICH compared to those with NON-OAC-ICH, with significant lower levels of EED in NOACs compared to NON-OAC-ICH. Increase of early PHE volume did not increase the likelihood of poor outcome in OAC-ICH but was independently associated with poor outcome in NON-OAC-ICH. The results underline the importance of etiology-specific treatment strategies. Further prospective studies are needed.

## Figures and Tables

**Figure 1 jcm-10-02234-f001:**
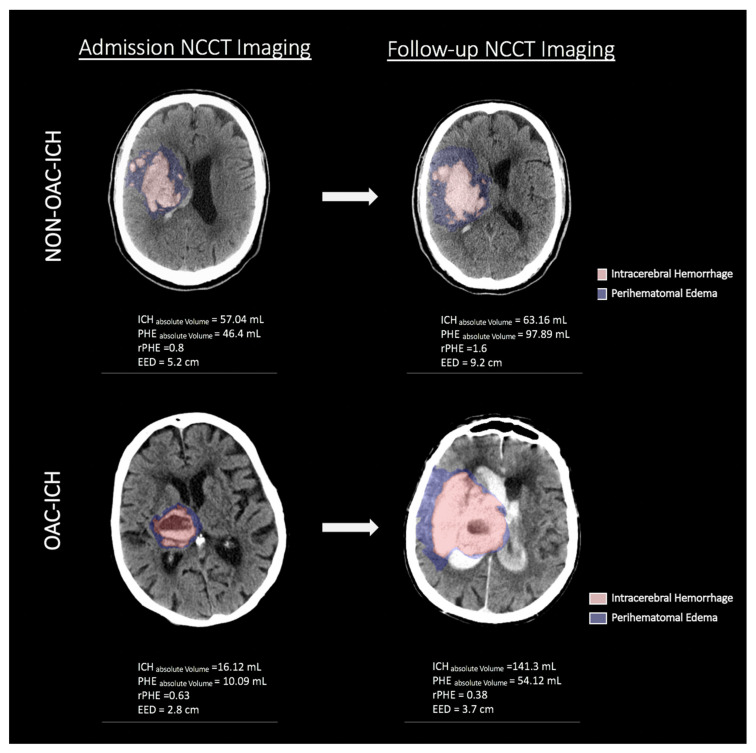
Illustrative examples of patients with spontaneous intracerebral hemorrhage (NON-OAC-ICH) and oral anticoagulation related intracerebral hemorrhage (OAC-ICH). Legend: ICH indicates intracerebral hemorrhage; EED, edema extension distance; NCCT, noncontrast computed tomography, OAC-ICH, OAC related intracerebral hemorrhage; PHE, perihematomal edema; rPHE, relative perihematomal edema; and NON-OAC-ICH, non-oral anticoagulation related intracerebral hemorrhage.

**Figure 2 jcm-10-02234-f002:**
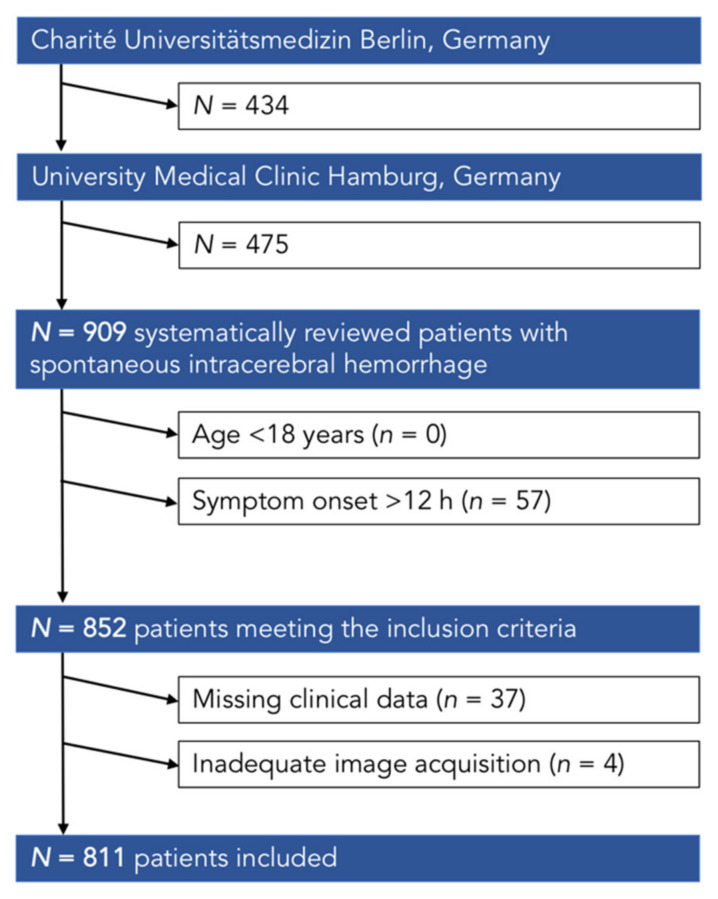
Patient flowchart.

**Table 1 jcm-10-02234-t001:** Baseline demographic and clinical characteristics of patients with non-oral anticoagulation associated intracerebral hemorrhage (NON-OAC-ICH) and oral anticoagulation associated intracerebral hemorrhage (OAC-ICH).

Baseline Characteristics	All (*n* = 811)	NON-OAC-ICH (*n* = 599)	OAC-ICH (*n* = 212)	*p*-value
Clinical parameters				
Age (years), median (IQR)	73 (60-79)	70 (58-77)	77 (70-82)	<0.001
Female, *n* (%)	348 (42.91)	254 (42.24)	94 (44.34)	0.61
Comorbidities				
• Hypertension, *n* (%)	624 (76.94)	447 (74.62)	177 (83.49)	0.008
• Diabetes mellitus, *n* (%)	120 (14.8)	80 (13.36)	40 (18.87)	0.048
Anticoagulation Treatment, *n* (%)	212 (26.14)	-	212 (100)	-
Antiplatelet Treatment, *n* (%)	190 (23.43)	160 (26.7)	30 (14.15)	<0.001
Time difference between symptom onset to imaging (hours), median (IQR)	2.81 (1.39–11.96)	2.81(1.28–12.57)	2.74 (1.56–11.65)	0.592
GCS admission, median (IQR)	12 (6–15)	12 (5–14)	11 (4–14)	0.315
NIHSS admission, median (IQR)	1 (8–14)	8 (1–15)	6 (1–14)	0.44
Radiological parameters				
ICH Volume (mL), mean (SD)	47 (54.11)	46.39 (53.85)	48.41 (54.84)	0.767
PHE Volume (mL), mean (SD)	36.42 (37.89)	37.61 (38.27)	33.06 (36.69)	0.091
rPHE, mean (SD)	1.2 (2.4)	1.25 (2.5)	1.08 (2.09)	0.001
EED, (cm), mean (SD)	4.57 (2.5)	4.72 (2.58)	4.14 (2.22)	0.007
Intraventricular hemorrhage, *n* (%)	374 (46.12)	279 (46.58)	95 (44.81)	0.799
ICH location, *n* (%)				
• lobar	362 (44.63)	261 (43.6)	101 (47.6)	0.317
• basal ganglia	306 (37.73)	238 (39.7)	68 (32.1)	0.05
• thalamic	16 (19.73)	11 (1.8)	5 (2.4)	0.636
• brainstem	37 (45.62)	27 (4.5)	10 (4.7)	0.896
• cerebellar	88 (10.85)	61 (10.2)	27 (12.7)	0.301
Procedure process				
Supratentorial craniectomy, *n* (%)	116 (14.3)	91 (15.19)	25 (11.79)	0.358
Suboccipital craniectomy, *n* (%)	33 (4.07)	20 (3.33)	13 (6.13)	0.85
EVD, *n* (%)	65 (8.01)	49 (8.2)	16 (7.55)	0.785
Clinical outcome				
mRS 90, *n* (%)				
• 0–3	225 (27.74)	164 (27.38)	61 (28.77)	0.24
• 4–6	586 (72.26)	435 (72.62)	151 (71.23)	0.7
• 6 (death)	205 (25.27)	147 (24.54)	58 (27.35)	0.068

Legend: ICH indicates intracerebral hemorrhage; IQR, interquartile range; IVH, intraventricular; GCS, Glasgow Come Scale; EED, edema extension distance; mRS, modified Rankin scale at 90 days; NIHSS, National Institutes of Health Stroke Scale; OAC-ICH, OAC associated intracerebral hemorrhage; PHE, perihematomal edema; ref, reference; and rPHE, relative perihematomal edema; NON-OAC-ICH, non-oral anticoagulation related intracerebral hemorrhage, EVD, extra-ventricular drainage and SD, standard deviation.

**Table 2 jcm-10-02234-t002:** Baseline demographic and clinical characteristics by patients with non-oral anticoagulation associated intracerebral hemorrhage (NON-OAC-ICH) and oral anticoagulation associated intracerebral hemorrhage with non-vitamin K antagonist oral anticoagulation (NOAC-ICH) and vitamin K antagonist associated intracerebral hemorrhage (VKA-ICH).

Radiological Parameters	NON-OAC-ICH (*n* = 599)	NOAC-ICH (*n* = 93)	VKA-ICH (*n* = 81)	*p*-value
ICH Volume (mL), mean (SD)	46.39 (53.84)	51.85 (66.43)	47.41 (43.7)	0.667
PHE Volume (mL), mean (SD)	37.61 (38.27)	32.94 (37.0)	36.21 (40.79)	0.544
rPHE, mean (SD)	1.2 (2.25)	1.11 (2.09)	0.88 (0.59)	0.399
EED, (cm), mean (SD)	4.72 (2.58)	4.07 (2.17)	4.34 (2.35)	0.042

Legend: ICH indicates intracerebral hemorrhage; EED, edema extension distance; PHE, perihematomal edema; rPHE, relative perihematomal edema; NOAC-ICH, non-vitamin K dependent oral anticoagulation associated intracerebral hemorrhage; NON-OAC-ICH, non-oral anticoagulation related intracerebral hemorrhage; SD, standard deviation; and VKA-ICH, vitamin K antagonists associated intracerebral hemorrhage. Please note: the number of patients (*n* = 599) displayed refers only to those with documented type of oral anticoagulation (*n* = 212 out of *n* = 811 were excluded).

**Table 3 jcm-10-02234-t003:** Multivariate analysis of predictors of poor outcome (modified Rankin scale 4–6 at 90 days) in patients with OAC associated intracerebral hemorrhage (OAC-ICH).

Predictor	Poor Outcome (mRS 4–6) in OAC-ICH			
	OR	95% CI	Regression- Coefficient	*p*-value
Age (years)	-	-	-	0.953
Gender (ref: female)	-	-	-	0.568
GCS on admission	0.758	0.629–0.912	−0.277	0.003
NIHSS on admission	1.123	1.028–1.227	0.116	0.10
PHE Volume (mL)	-	-	-	0.451
IVH (ref: yes)	-	-	-	0.213
Craniectomy (ref: supratentorial)	-	-	-	0.998

Legend: Multivariate logistic regression analysis for independent outcome variables of clinical outcome, modified Rankin scale poor outcome, mRS 4–6 at 90 days. Note: non-significant independent variables are only displayed with *p*-values (gender). Collinear covariates, as expressed by a variance inflation factor > 5, were identified, and 1 covariate was removed from the model (ICH volume). ICH indicates intracerebral hemorrhage; IVH, intraventricular; GCS, Glasgow Coma Scale; and OAC-ICH, OAC associated intracerebral hemorrhage; and PHE, perihematomal edema; ref, reference.

**Table 4 jcm-10-02234-t004:** Multivariate analysis of predictors of poor outcome (modified Rankin scale 4–6 at 90 days) in patients with non-oral anticoagulation associated intracerebral hemorrhage (NON-OAC-ICH).

Predictor	Poor Outcome (mRS 4–6) in NON-OAC-ICH			
	OR	95% CI	Regression- Coefficient	*p*-value
Age (years)	-	-	-	0.248
Gender (ref: female)	0.520	0.283–0.955	−0.654	0.035
GCS on admission	0.758	0.676–0.849	−0.277	<0.0001
NIHSS on admission	1.142	1.083–1.204	0.132	<0.0001
PHE Volume (mL)	1.015	1.002–1.028	0.014	0.027
EED (cm)	-	-	-	0.843
IVH (ref: yes)	2.803	1.441–5.451	1.031	0.002
Location (ref: supratentorial)	0.381	0.201–0.723	−0.966	0.003
Craniectomy (ref: supratentorial)	-	-	-	0.113

Legend: Multivariate logistic regression analysis for independent outcome variables of clinical outcome, modified Rankin scale poor outcome, mRS 4–6 at 90 days. Note: non-significant independent variables are only displayed with *p*-values (gender). Collinear covariates, as expressed by a variance inflation factor >5, were identified, and 1 covariate was removed from the model (ICH volume and EED). ICH indicates intracerebral hemorrhage; IVH, intraventricular; EED, edema extension distance; GCS; Glasgow Coma Scale; and PHE, perihematomal edema; ref, reference; and NON-OAC-ICH, non-oral anticoagulation related intracerebral hemorrhage.

## Data Availability

The data that support the findings of this study are available from the corresponding authors upon reasonable request and accordance with the institution’s data security regulations.

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
