# Peer review of "Perihematomal Edema and Clinical Outcome in Intracerebral Hemorrhage Related to Different Oral Anticoagulants"

_jcm, 2021, doi:10.3390/jcm10112234_

Round 1
Reviewer 1 Report
Every comments have been rightly addressed and I have no more comments.
Author Response
We thank the Editors and Referees for their interest in our work and for the helpful comments that have greatly improved the manuscript. We appreciate the opportunity to clarify our research objectives and results.
Reviewer #1:
Every comments have been rightly addressed and I have no more comments.
Re: We thank you the reviewer for this positive evaluation and appreciate the comments which have helped us to improve the overall quality of the manuscript.

Reviewer 2 Report
In the present study, the authors present a retrospective analysis of 811 patients with spontaneous ICH. The aim of this study is to examine the effects of different types of oral anticoagulants on perihematomal edema (PHE) as a potential biomarker for secondary brain injury and clinical outcome. Relative PHE (rPHE), Edema Expansion Distance (EED) were calculated for patients with OAC-ICH and NON-OAC-ICH. Mean rPHE and mean EED were significantly lower in patients with OAC-ICH compared to NON-OAC-ICH although whereas mean absolute PHE did not differ. As conclusion the authors mention that early increase of PHE volume did not increase the likelihood of poor outcome in OAC-ICH but was independentlyassociated withpoor outcome in NON-OAC-ICH.
The manuscript deals with an important issue: the influence of oral anticoagulants on ICH. Information on the influence of oral anticoagulant s on the course of ICH are scare and urgently needed in daily routine. The manuscript is well structured to facilitate good readability and comprehension. The included figures are suited to convey the presented information. The text is fluent.
Comments:
- Hematoma volume and PHE did not differ between patients with and without OAC. On a more detailed look, the authors found that mean rPHE and mean EED were significantly lower in patients with OAC-ICH. Although still very interesting, the clinical relevance of this finding has to be further substantiated.
- The manuscript is well written with only minor speeling errors
Author Response
We thank the Editors and Referees for their interest in our work and for the helpful comments that will greatly improve the manuscript. We have tried to do our best to respond to the points raised and we appreciate the opportunity to clarify our research objectives and results.
As indicated below, we have checked all the general and specific comments provided by the Referees and have made necessary changes accordingly to their indications.

Reviewer 3 Report
The Authors present the the effects of different oral anticoagulant therapies associated with intracerebral hemorrhage on perilesional edema.
The retrospective study includes 811 cases of which 212 had OAC-ICH. The results demonstrate how the quantitative markers of early perihematomal edema (PHE) were lower in OAC-ICH than in NO OAC-ICH goup; and conclude the increase of early PHE did not increase the chance of poor outcome in OAC-ICH patients . The Authors have done a good work; with the anaqlysis and study design. The conclusions are sound and limitations correctly stated .
Author Response
We thank the Editors and Referees for their interest in our work and overall positive evaluation of the manuscript.
Reviewer #3:
The Authors present the effects of different oral anticoagulant therapies associated with intracerebral hemorrhage on perilesional edema.
The retrospective study includes 811 cases of which 212 had OAC-ICH. The results demonstrate how the quantitative markers of early perihematomal edema (PHE) were lower in OAC-ICH than in NO OAC-ICH goup; and conclude the increase of early PHE did not increase the chance of poor outcome in OAC-ICH patients. The Authors have done a good work; with the analysis and study design. The conclusions are sound and limitations correctly stated.
Re: We thank Reviewer 3 for the positive evaluation of our manuscript and very much appreciate the reviewer’s time and effort to review our work, thank you.
This manuscript is a resubmission of an earlier submission. The following is a list of the peer review reports and author responses from that submission.
Round 1
Reviewer 1 Report
It is a well written and interesting study which increases the evidence in a difficult topic: how to treat ICH related to OAC.
I have some minor comments:
-Introduction: I think it is too long and it should be focused. There is a wongly stopped sentence: medical treatment options and.
-Results: The authors said that EED was an independent variable associated to poor outcome in NON-OAC-ICH but afterwards they said that EED did not increase the likelihood for poor outcome (it is difficult to explain).
-Discussion: Many studies have associated OAC with poor clinical outcome. I think that authors should discuss the reason of the absence of association in this study.
Author Response
We thank you the reviewer for his or her overall positive evaluation and appreciate the suggestions and comments, which we have tried to pursue by additional revisions that are described in this revised manuscript. We have addressed each of the comments point by point in the following and have added all changes in the manuscript accordingly. Please see the file attached.

Reviewer 2 Report
Overall, the present manuscript investigates peri-hematomal edema formation comapring OAC-ICH to spontaneous ICH in a fairly large cohort. The topic is of certain interest and currently limited clinical data is available.
Yet several shortcommings need to be adressed.
- Statistical analyses does not support the presented conclusions.
- Please include ICH volume into the models
- Pls check for co-linear variables
- Importantly timing of follow-imaging needs to be considered
- Exclude surgically treated patients from all analyses
- No statement has been made regarding the time interval for edema assessment, which represents a strong confounder of edema extent.
- Adjusting for ICH volume upon mv-models is requiste, as this represents the strongest factor influencing rel. edema
- Please present max. edema, assessed when?
- relative edema was calculated at what time of FU?
- How was ICH volume determined? Methodology. Presented ICH volume appears very large in relation to neurological admission status.
- Were different imaging modalities corrected for?
- Patients who received craniectomy need to be excluded from the entire analyses.
Author Response
We thank Reviewer 2 for the evaluation of our manuscript. We very much appreciate the suggestions and comments, which we have tried to pursue by additional revisions that are described in this revised manuscript. We have addressed each of the comments point by point in the following and have added all changes in the manuscript accordingly. Please see the file attached.
